# A Pseudo-Euclidean Iteration for Optimal Recovery in Noisy ICA

**James Voss**
The Ohio State University
vossj@cse.ohio-state.edu

**Mikhail Belkin**
The Ohio State University
mbelkin@cse.ohio-state.edu

**Luis Rademacher**
The Ohio State University
lrademac@cse.ohio-state.edu

## Abstract

Independent Component Analysis (ICA) is a popular model for blind signal separation. The ICA model assumes that a number of independent source signals are linearly mixed to form the observed signals. We propose a new algorithm, PEGI (for pseudo-Euclidean Gradient Iteration), for provable model recovery for ICA with Gaussian noise. The main technical innovation of the algorithm is to use a fixed point iteration in a pseudo-Euclidean (indefinite "inner product") space. The use of this indefinite "inner product" resolves technical issues common to several existing algorithms for noisy ICA. This leads to an algorithm which is conceptually simple, efficient and accurate in testing.

Our second contribution is combining PEGI with the analysis of objectives for optimal recovery in the noisy ICA model. It has been observed that the direct approach of demixing with the inverse of the mixing matrix is suboptimal for signal recovery in terms of the natural Signal to Interference plus Noise Ratio (SINR) criterion. There have been several partial solutions proposed in the ICA literature. It turns out that any solution to the mixing matrix reconstruction problem can be used to construct an SINR-optimal ICA demixing, despite the fact that SINR itself cannot be computed from data. That allows us to obtain a practical and provably SINR-optimal recovery method for ICA with arbitrary Gaussian noise.

## 1   Introduction

Independent Component Analysis refers to a class of methods aiming at recovering statistically independent signals by observing their unknown linear combination. There is an extensive literature on this and a number of related problems [7].

In the ICA model, we observe $n$-dimensional realizations $\mathbf{x}(1), \ldots, \mathbf{x}(N)$ of a latent variable model $\mathbf{X} = \sum_{k=1}^{m} S_k A_k = A\mathbf{S}$ where $A_k$ denotes the $k^{\text{th}}$ column of the $n \times m$ mixing matrix $A$ and $\mathbf{S} = (S_1, \ldots, S_m)^T$ is the unseen latent random vector of "signals". It is assumed that $S_1, \ldots, S_m$ are independent and non-Gaussian. The source signals and entries of $A$ may be either real- or complex-valued. For simplicity, we will assume throughout that $\mathbf{S}$ has zero mean, as this may be achieved in practice by centering the observed data.

Many ICA algorithms use the preprocessing "whitening" step whose goal is to orthogonalize the independent components. In the noiseless, case this is commonly done by computing the square root of the covariance matrix of $\mathbf{X}$. Consider now the noisy ICA model $\mathbf{X} = A\mathbf{S} + \boldsymbol{\eta}$ with additive $\mathbf{0}$-mean noise $\boldsymbol{\eta}$ independent of $\mathbf{S}$. It turns out that the introduction of noise makes accurate recovery of the signals significantly more involved. Specifically, whitening using the covariance matrix does not work in the noisy ICA model as the covariance matrix combines both signal and noise. For the case when the noise is Gaussian, matrices constructed from higher order statistics (specifically, cumulants) can be used instead of the covariance matrix. However, these matrices are not in general positive definite and thus the square root cannot always be extracted. This limits the applicability of

several previous methods, such as [1, 2, 9]. The GI-ICA algorithm proposed in [21] addresses this issue by using a complicated quasi-orthogonalization step followed by an iterative method.

In this paper (section 2), we develop a simple and practical one-step algorithm, PEGI (for pseudo-Euclidean Gradient Iteration) for provably recovering $A$ (up to the unavoidable ambiguities of the model) in the case when the noise is Gaussian (with an arbitrary, unknown covariance matrix). The main technical innovation of our approach is to formulate the recovery problem as a fixed point method in an indefinite (pseudo-Euclidean) "inner product" space.

The second contribution of the paper is combining PEGI with the analysis of objectives for optimal recovery in the noisy ICA model. In most applications of ICA (e.g., speech separation [18], MEG/EEG artifact removal [20] and others) one cares about recovering the signals $\mathbf{s}(1), \ldots, \mathbf{s}(N)$. This is known as the *source recovery problem*. This is typically done by first recovering the matrix $A$ (up to an appropriate scaling of the column directions).

At first, source recovery and recovering the mixing matrix $A$ appear to be essentially equivalent. In the *noiseless* ICA model, if $A$ in invertible[1] then $\mathbf{s}(t) = A^{-1}\mathbf{x}(t)$ recovers the sources. On the other hand, in the *noisy* model, the exact recovery of the latent sources $\mathbf{s}(t)$ becomes impossible even if $A$ is known exactly. Part of the "noise" can be incorporated into the "signal" preserving the form of the model. Even worse, neither $A$ nor $\mathbf{S}$ are defined uniquely as there is an inherent ambiguity in the setting. There could be many equivalent decompositions of the observed signal as $\mathbf{X} = A'\mathbf{S}' + \boldsymbol{\eta}'$ (see the discussion in section 3).

We consider recovered signals of the form $\hat{\mathbf{S}}(B) := B\mathbf{X}$ for a choice of $m \times n$ demixing matrix $B$. Signal recovery is considered optimal if the coordinates of $\hat{\mathbf{S}}(B) = (\hat{S}_1(B), \ldots, \hat{S}_m(B))$ maximize Signal to Interference plus Noise Ratio (SINR) within any *fixed* model $\mathbf{X} = A\mathbf{S} + \boldsymbol{\eta}$. Note that the value of SINR depends on the decomposition of the observed data into "noise" and "signal": $\mathbf{X} = A'\mathbf{S}' + \boldsymbol{\eta}'$.

Surprisingly, the SINR optimal demixing matrix does not depend on the decomposition of data into signal plus noise. As such, SINR optimal ICA recovery is well defined given access to data despite the inherent ambiguity in the model. Further, it will be seen that the SINR optimal demixing can be constructed from $\operatorname{cov}(\mathbf{X})$ and the directions of the columns of $A$ (which are also well-defined across signal/noise decompositions).

Our SINR-optimal demixing approach combined with the PEGI algorithm provides a complete SINR-optimal recovery algorithm in the ICA model with arbitrary Gaussian noise. We note that the ICA papers of which we are aware that discuss optimal demixing do not observe that SINR optimal demixing is invariant to the choice of signal/noise decomposition. Instead, they propose more limited strategies for improving the demixing quality within a fixed ICA model. For instance, Joho et al. [14] show how SINR-optimal demixing can be approximated with extra sensors when assuming a white additive noise, and Koldovskỳ and Tichavskỳ [16] discuss how to achieve asymptotically low bias ICA demixing assuming white noise within a fixed ICA model. However, the invariance of the SINR-optimal demixing matrix appears in the array sensor systems literature [6].

Finally, in section 4, we demonstrate experimentally that our proposed algorithm for ICA outperforms existing practical algorithms at the task of noisy signal recovery, including those specifically designed for beamforming, when given sufficiently many samples. Moreover, most existing practical algorithms for noisy source recovery have a bias and cannot recover the optimal demixing matrix even with infinite samples. We also show that PEGI requires significantly fewer samples than GI-ICA [21] to perform ICA accurately.

## 1.1 The Indeterminacies of ICA

**Notation:** We use $M^*$ to denote the entry-wise complex conjugate of a matrix $M$, $M^T$ to denote its transpose, $M^H$ to denote its conjugate transpose, and $M^\dagger$ to denote its Moore-Penrose pseudoinverse.

Before proceeding with our results, we discuss the somewhat subtle issue of indeterminacies in ICA. These ambiguities arise from the fact that the observed $\mathbf{X}$ may have multiple decompositions into ICA models $\mathbf{X} = A\mathbf{S} + \boldsymbol{\eta}$ and $\mathbf{X} = A'\mathbf{S}' + \boldsymbol{\eta}'$.

Noise-free ICA has two natural indeterminacies. For any nonzero constant $\alpha$, the contribution of the $k^{\text{th}}$ component $A_k S_k$ to the model can equivalently be obtained by replacing $A_k$ with $\alpha A_k$ and $S_k$ with the rescaled signal $\frac{1}{\alpha} S_k$. To lessen this scaling indeterminacy, we use the convention[2] that $\text{cov}(\mathbf{S}) = \mathcal{I}$ throughout this paper. As such, each source $S_k$ (or equivalently each $A_k$) is defined up to a choice of sign (a unit modulus factor in the complex case). In addition, there is an ambiguity in the order of the latent signals. For any permutation $\pi$ of $[m]$ (where $[m] := \{1, \ldots, m\}$), the ICA models $\mathbf{X} = \sum_{k=1}^{m} S_k A_k$ and $\mathbf{X} = \sum_{k=1}^{m} S_{\pi(k)} A_{\pi(k)}$ are indistinguishable. In the noise free setting, $A$ is said to be recovered if we recover each column of $A$ up to a choice of sign (or up to a unit modulus factor in the complex case) and an unknown permutation. As the sources $S_1, \ldots, S_m$ are only defined up to the same indeterminacies, inverting the recovered matrix $\tilde{A}$ to obtain a demixing matrix works for signal recovery.

In the noisy ICA setting, there is an additional indeterminacy in the definition of the sources. Consider a $\mathbf{0}$-mean axis-aligned Gaussian random vector $\boldsymbol{\xi}$. Then, the noisy ICA model $\mathbf{X} = A(\mathbf{S} + \boldsymbol{\xi}) + \boldsymbol{\eta}$ in which $\boldsymbol{\xi}$ is considered part of the latent source signal $\mathbf{S}' = \mathbf{S} + \boldsymbol{\xi}$, and the model $\mathbf{X} = A\mathbf{S} + (A\boldsymbol{\xi} + \boldsymbol{\eta})$ in which $\boldsymbol{\xi}$ is part of the noise are indistinguishable. In particular, the latent source $\mathbf{S}$ and its covariance are ill-defined. Due to this extra indeterminacy, the lengths of the columns of $A$ no longer have a fully defined meaning even when we assume $\text{cov}(\mathbf{S}) = \mathcal{I}$. In the noisy setting, $A$ is said to be recovered if we obtain the columns of $A$ up to non-zero scalar multiplicative factors and an arbitrary permutation.

The last indeterminacy is the most troubling as it suggests that the power of each source signal is itself ill-defined in the noisy setting. Despite this indeterminacy, it is possible to perform an SINR-optimal demixing without additional assumptions about what portion of the signal is source and what portion is noise. In section 3, we will see that SINR-optimal source recovery takes on a simple form: Given any solution $\tilde{A}$ which recovers $A$ up to the inherent ambiguities of noisy ICA, then $\tilde{A}^H \text{cov}(\mathbf{X})^\dagger$ is an SINR-optimal demixing matrix.

## 1.2 Related Work and Contributions

Independent Component Analysis is probably the most used model for Blind Signal Separation. It has seen numerous applications and has generated a vast literature, including in the noisy and underdetermined settings. We refer the reader to the books [7, 13] for a broad overview of the subject.

It was observed early on by Cardoso [4] that ICA algorithms based soley on higher order cumulant statistics are invariant to additive Gaussian noise. This observation has allowed the creation of many algorithms for recovering the ICA mixing matrix in the noisy and often underdetermined settings. Despite the significant work on noisy ICA algorithms, they remain less efficient, more specialized, or less practical than the most popular noise free ICA algorithms.

Research on cumulant-based noisy ICA can largely be split into several lines of work which we only highlight here. Some algorithms such as FOOBI [4] and BIOME [1] directly use the tensor structure of higher order cumulants. In another line of work, De Lathauwer et al. [8] and Yeredor [23] have suggested algorithms which jointly diagonalize cumulant matrices in a manner reminiscent of the noise-free JADE algorithm [3]. In addition, Yeredor [22] and Goyal et al. [11] have proposed ICA algorithms based on random directional derivatives of the second characteristic function.

Each line of work has its advantages and disadvantages. The joint diagonalization algorithms and the tensor based algorithms tend to be practical in the sense that they use redundant cumulant information in order to achieve more accurate results. However, they have a higher memory complexity than popular noise free ICA algorithms such as FastICA [12]. While the tensor methods (FOOBI and BIOME) can be used when there are more sources than the dimensionality of the space (the underdetermined ICA setting), they require all the latent source signals to have positive order $2k$ cumulants ($k \geq 2$, a predetermined fixed integer) as they rely on taking a matrix square root. Finally, the methods based on random directional derivatives of the second characteristic function rely heavily upon randomness in a manner not required by the most popular noise free ICA algorithms.

We continue a line of research started by Arora et al. [2] and Voss et al. [21] on fully determined noisy ICA which addresses some of these practical issues by using a deflationary approach reminiscent of FastICA. Their algorithms thus have lower memory complexity and are more scalable to high dimensional data than the joint diagonalization and tensor methods. However, both works require

a preprocessing step (quasi-orthogonalization) to orthogonalize the latent signals which is based on taking a matrix square root. Arora et al. [2] require each latent signal to have positive fourth cumulant in order to carry out this preprocessing step. In contrast, Voss et al. [21] are able to perform quasi-orthogonalization with source signals of mixed sign fourth cumulants; but their quase-orthogonalization step is more complicated and can run into numerical issues under sampling error. We demonstrate that quasi-orthogonalization is unnecessary. We introduce the PEGI algorithm to work within a (not necessarily positive definite) inner product space instead. Experimentally, this leads to improved demixing performance. In addition, we handle the case of complex signals.

Finally, another line of work attempts to perform SINR-optimal source recovery in the noisy ICA setting. It was noted by Koldovskỳ and Tichavskỳ [15] that for noisy ICA, traditional ICA algorithms such as FastICA and JADE actually outperform algorithms which first recover $A$ in the noisy setting and then use the resulting approximation of $A^\dagger$ to perform demixing. It was further observed that $A^\dagger$ is not the optimal demixing matrix for source recovery. Later, Koldovskỳ and Tichavskỳ [17] proposed an algorithm based on FastICA which performs a low SINR-bias beamforming.

## 2  Pseudo-Euclidean Gradient Iteration ICA

In this section, we introduce the PEGI algorithm for recovering $A$ in the "fully determined" noisy ICA setting where $m \leq n$. PEGI relies on the idea of Gradient Iteration introduced Voss et al. [21]. However, unlike GI-ICA Voss et al. [21], PEGI does not require the source signals to be orthogonalized. As such, PEGI does not require the complicated quasi-orthogonalization preprocessing step of GI-ICA which can be inaccurate to compute in practice. We sketch the Gradient Iteration algorithm in Section 2.1, and then introduce PEGI in Section 2.2. For simplicity, we limit this discussion to the case of real-valued signals. A mild variation of our PEGI algorithm works for complex-valued signals, and its construction is provided in the supplementary material.

In this section we assume a noisy ICA model $\mathbf{X} = A\mathbf{S} + \boldsymbol{\eta}$ such that $\boldsymbol{\eta}$ is arbitrary Gaussian and independent of $\mathbf{S}$. We also assume that $m \leq n$, that $m$ is known, and that the columns of $A$ are linearly independent.

### 2.1  Gradient Iteration with Orthogonality

The gradient iteration relies on the properties of cumulants. We will focus on the fourth cumulant, though similar constructions may be given using other even order cumulants of higher order. For a zero-mean random variable $X$, the fourth order cumulant may be defined as $\kappa_4(X) := \mathbb{E}[X^4] - 3\mathbb{E}[X^2]^2$ [see 7, Chapter 5, Section 1.2]. Higher order cumulants have nice algebraic properties which make them useful for ICA. In particular, $\kappa_4$ has the following properties: (1) (Independence) If $X$ and $Y$ are independent, then $\kappa_4(X + Y) = \kappa_4(X) + \kappa_4(Y)$. (2) (Homogeneity) If $\alpha$ is a scalar, then $\kappa_4(\alpha X) = \alpha^4 \kappa_4(X)$. (3) (Vanishing Gaussians) If $X$ is normally distributed then $\kappa_4(X) = 0$.

We consider the following function defined on the unit sphere: $f(\mathbf{u}) := \kappa_4(\langle \mathbf{X}, \mathbf{u} \rangle)$. Expanding $f(\mathbf{u})$ using the above properties we obtain:

$$f(\mathbf{u}) = \kappa_4\big(\sum\nolimits_{k=1}^m \langle A_k, \mathbf{u} \rangle S_k + \langle \mathbf{u}, \boldsymbol{\eta} \rangle\big) = \sum\nolimits_{k=1}^m \langle A_k, \mathbf{u} \rangle^4 \kappa_4(S_k) \,.$$

Taking derivatives we obtain:

$$\nabla f(\mathbf{u}) = 4\sum\nolimits_{k=1}^m \langle A_k, \mathbf{u} \rangle^3 \kappa_4(S_k) A_k \tag{1}$$

$$\mathcal{H}f(\mathbf{u}) = 12\sum\nolimits_{k=1}^m \langle A_k, \mathbf{u} \rangle^2 \kappa_4(S_k) A_k A_k^T = AD(\mathbf{u})A^T \tag{2}$$

where $D(\mathbf{u})$ is a diagonal matrix with entries $D(\mathbf{u})_{kk} = 12\langle A_k, \mathbf{u} \rangle^2 \kappa_4(S_k)$. We also note that $f(\mathbf{u})$, $\nabla f(\mathbf{u})$, and $\mathcal{H}f(\mathbf{u})$ have natural sample estimates (see [21]).

Voss et al. [21] introduced GI-ICA as a fixed point algorithm under the assumption that the columns of $A$ are orthogonal but not necessarily unit vectors. The main idea is that the update $\mathbf{u} \leftarrow \nabla f(\mathbf{u})/\|\nabla f(\mathbf{u})\|$ is a form of a generalized power iteration. From equation (1), each $A_k$ may be considered as a direction in a hidden orthogonal basis of the space. During each iteration, the $A_k$ coordinate of $\mathbf{u}$ is raised to the 3$^{\text{rd}}$ power and multiplied by a constant. Treating this iteration as a fixed point update, it was shown that given a random starting point, this iterative procedure converges rapidly to one of the columns of $A$ (up to a choice of sign). The rate of convergence is cubic.

However, the GI-ICA algorithm requires a somewhat complicated preprocessing step called quasi-orthogonalization to linearly transform the data to make columns of $A$ orthogonal. Quasi-orthogonalization makes use of evaluations of Hessians of the fourth cumulant function to construct a matrix of the form $C = ADA^T$ where $D$ has all positive diagonal entries—a task which is complicated by the possibility that the latent signals $S_i$ may have fourth order cumulants of differing signs—and requires taking the matrix square root of a positive definite matrix of this form. However, the algorithm used for constructing $C$ under sampling error is not always positive definite in practice, which can make the preprocessing step fail. We will show how our PEGI algorithm makes quasi-orthogonalization unnecessary, in particular, resolving this issue.

## 2.2 Gradient Iteration in a Pseudo-Euclidean Space

We now show that the gradient iteration can be performed using in a pseudo-Euclidean space in which the columns of $A$ are orthogonal. The natural candidate for the "inner product space" would be to use $\langle \cdot, \cdot \rangle_*$ defined as $\langle \mathbf{u}, \mathbf{v} \rangle_* := \mathbf{u}^T (AA^T)^\dagger \mathbf{v}$. Clearly, $\langle A_i, A_j \rangle_* = \delta_{ij}$ gives the desired orthogonality property. However, there are two issues with this "inner product space": First, it is only an inner product space when $A$ is invertible. This turns out not to be a major issue, and we move forward largely ignoring this point. The second issue is more fundamental: We only have access to $AA^T$ in the noise free setting where $\text{cov}(\mathbf{X}) = AA^T$. In the noisy setting, we have access to matrices of the form $\mathcal{H}f(\mathbf{u}) = AD(\mathbf{u})A^T$ from equation (2) instead.

**Algorithm 1** Recovers a column of $A$ up to a scaling factor if $\mathbf{u}_0$ is generically chosen.

> **Inputs:** Unit vector $\mathbf{u}_0$, $C$, $\nabla f$
> $k \leftarrow 1$
> **repeat**
> $\quad \mathbf{u}_k \leftarrow \nabla f(C^\dagger \mathbf{u}_{k-1})/\|\nabla f(C^\dagger \mathbf{u}_{k-1})\|$
> $\quad k \leftarrow k + 1$
> **until** Convergence (up to sign)
> **return** $\mathbf{u}_k$

We consider a pseudo-Euclidean inner product defined as follows: Let $C = ADA^T$ where $D$ is a diagonal matrix with non-zero diagonal entries, and define $\langle \cdot, \cdot \rangle_C$ by $\langle \mathbf{u}, \mathbf{v} \rangle_C = \mathbf{u}^T C^\dagger \mathbf{v}$. When $D$ contains negative entries, this is not a proper inner product since $C$ is not positive definite. In particular, $\langle A_k, A_k \rangle_C = A_k^T(ADA^T)^\dagger A_k = d_{kk}^{-1}$ may be negative. Nevertheless, when $k \neq j$, $\langle A_k, A_j \rangle_C = A_k^T(ADA^T)^\dagger A_j = 0$ gives that the columns of $A$ are orthogonal in this space.

We define functions $\alpha_k : \mathbb{R}^n \to \mathbb{R}$ by $\alpha_k(\mathbf{u}) = (A^\dagger \mathbf{u})_k$ such that for any $\mathbf{u} \in \text{span}(A_1, \ldots, A_m)$, then $\mathbf{u} = \sum_{i=1}^m \alpha_i(\mathbf{u})A_i$ is the expansion of $\mathbf{u}$ in its $A_i$ basis. Continuing from equation (1), for any $\mathbf{u} \in S^{n-1}$ we see $\nabla f(C^\dagger \mathbf{u}) = 4 \sum_{k=1}^n \langle A_k, C^\dagger \mathbf{u} \rangle^3 \kappa_4(S_k)A_k = 4 \sum_{k=1}^n \langle A_k, \mathbf{u} \rangle_C^3 \kappa_4(S_k)A_k$ is the gradient iteration recast in the $\langle \cdot, \cdot \rangle_C$ space. Expanding $\mathbf{u}$ in its $A_k$ basis, we obtain

$$\nabla f(C^\dagger \mathbf{u}) = 4 \sum_{k=1}^m (\alpha_k(\mathbf{u})\langle A_k, A_k \rangle_C)^3 \kappa_4(S_k)A_k = 4 \sum_{k=1}^m \alpha_k(\mathbf{u})^3(d_{kk}^{-3}\kappa_4(S_k))A_k , \quad (3)$$

which is a power iteration in the unseen $A_k$ coordinate system. As no assumptions are made upon the $\kappa_4(S_k)$ values, the $d_{kk}^{-3}$ scalings which were not present in eq. (1) cause no issues. Using this update, we obtain Alg. 1, a fixed point method for recovering a single column of $A$ up to an unknown scaling.

Before proceeding, we should clarify the notion of fixed point convergence in Algorithm 1. We say that the sequence $\{\mathbf{u}_k\}_{k=0}^\infty$ converges to $\mathbf{v}$ up to sign if there exists a sequence $\{c_k\}_{k=0}^\infty$ such that each $c_k \in \{\pm 1\}$ and $c_k \mathbf{u}_k \to \mathbf{v}$ as $k \to \infty$. We have the following convergence guarantee.

**Theorem 1.** *If $\mathbf{u}_0$ is chosen uniformly at random from $S^{n-1}$, then with probability 1, there exists $\ell \in [m]$ such that the sequence $\{\mathbf{u}_k\}_{k=0}^\infty$ defined as in Algorithm 1 converges to $A_\ell/\|A_\ell\|$ up to sign. Further, the rate of convergence is cubic.*

Due to limited space, we omit the proof of Theorem 1. It is similar to the proof of [21, Theorem 4].

In practice, we test near convergence by checking if we are still making significant progress. In particular, for some predefined $\epsilon > 0$, if there exists a sign value $c_k \in \{\pm 1\}$ such that $\|\mathbf{u}_k - c_k \mathbf{u}_{k-1}\| < \epsilon$, then we declare convergence achieved and return the result. As there are only two choices for $c_k$, this is easily checked, and we exit the loop if this condition is met.

**Full ICA Recovery Via the Pseudo-Euclidean GI-Update.** We are able to recover a single column of $A$ up to its unknown scale. However, for full recovery of $A$, we would like (given recovered columns $A_{\ell_1}, \ldots, A_{\ell_j}$) to be able to recover a column $A_k$ such that $k \notin \{\ell_1, \ldots, \ell_j\}$ on demand.

The idea behind the simultaneous recovery of all columns of $A$ is two-fold. First, instead of just finding columns of $A$ using Algorithm 1, we simultaneously find rows of $A^\dagger$. Then, using the

recovered columns of $A$ and rows of $A^\dagger$, we project $\mathbf{u}$ onto the orthogonal complement of the recovered columns of $A$ within the $\langle \cdot, \cdot \rangle_C$ pseudo-inner product space.

**Recovering rows of $A^\dagger$.** Suppose we have access to a column $A_k$ (which may be achieved using Algorithm 1). Let $A_{k\cdot}^\dagger$ denote the $k^{\text{th}}$ row of $A^\dagger$. Then, we note that $C^\dagger A_k = (ADA^T)^\dagger A_k = d_{kk}^{-1}(A^T)_k^\dagger = d_{kk}^{-1}(A_{k\cdot}^\dagger)^T$ recovers $A_{k\cdot}^\dagger$ up to an arbitrary, unknown constant $d_{kk}^{-1}$. However, the constant $d_{kk}^{-1}$ may be recovered by noting that $\langle A_k, A_k \rangle_C = (C^\dagger A_k)^T A_k = d_{kk}^{-1}$. As such, we may estimate $A_{k\cdot}^\dagger$ as $[C^\dagger A_k/((C^\dagger A_k)^T A_k)]^T$.

---

**Algorithm 2** Full ICA matrix recovery algorithm. Returns two matrices: (1) $\tilde{A}$ is the recovered mixing matrix for the noisy ICA model $\mathbf{X} = A\mathbf{S} + \boldsymbol{\eta}$, and (2) $\tilde{B}$ is a running estimate of $\tilde{A}^\dagger$.

---

1: **Inputs:** $C, \nabla f$
2: $\tilde{A} \leftarrow 0, \tilde{B} \leftarrow 0$
3: **for** $j \leftarrow 1$ to $m$ **do**
4:     Draw $\mathbf{u}$ uniformly at random from $S^{n-1}$.
5:     **repeat**
6:         $\mathbf{u} \leftarrow \mathbf{u} - \tilde{A}\tilde{B}\mathbf{u}$
7:         $\mathbf{u} \leftarrow \nabla f(C^\dagger \mathbf{u})/\|\nabla f(C^\dagger \mathbf{u})\|$.
8:     **until** Convergence (up to sign)
9:     $\tilde{A}_j \leftarrow \mathbf{u}$
10:    $\tilde{B}_{j\cdot} \leftarrow [C^\dagger A_j/((C^\dagger A_j)^T A_j)]^T$
11: **end for**
12: **return** $\tilde{A}, \tilde{B}$

---

**Enforcing Orthogonality During the GI Update.** Given access to a vector $\mathbf{u} = \sum_{k=1}^m \alpha_k(\mathbf{u})A_k + P_{A\perp}\mathbf{u}$ (where $P_{A\perp}$ is the projection onto the orthogonal complements of the range of $A$), some recovered columns $A_{\ell_1}, \ldots, A_{\ell_r}$, and corresponding rows of $A^\dagger$, we may zero out the components of $\mathbf{u}$ corresponding to the recovered columns of $A$. Letting $\mathbf{u}' = \mathbf{u} - \sum_{j=1}^r A_{\ell_j} A_{\ell_j\cdot}^\dagger \mathbf{u}$, then $\mathbf{u}' = \sum_{k \in [m]\setminus\{\ell_1,\ldots,\ell_r\}} \alpha_k(\mathbf{u})A_k + P_{A\perp}\mathbf{u}$. In particular, $\mathbf{u}'$ is orthogonal (in the $\langle \cdot, \cdot \rangle_C$ space) to the previously recovered columns of $A$. This allows the non-orthogonal gradient iteration algorithm to recover a new column of $A$.

Using these ideas, we obtain Algorithm 2, which is the PEGI algorithm for recovery of the mixing matrix $A$ in noisy ICA up to the inherent ambiguities of the problem. Within this Algorithm, step 6 enforces orthogonality with previously found columns of $A$, guaranteeing that convergence to a new column of $A$.

**Practical Construction of $C$.** In our implementation, we set $C = \frac{1}{12}\sum_{k=1}^n \mathcal{H}f(\mathbf{e}_k)$, as it can be shown from equation (2) that $\sum_{k=1}^n \mathcal{H}f(\mathbf{e}_k) = ADA^T$ with $d_{kk} = \|A_k\|^2 \kappa_4(S_k)$. This deterministically guarantees that each latent signal has a significant contribution to $C$.

## 3   SINR Optimal Recovery in Noisy ICA

In this section, we demonstrate how to perform SINR optimal ICA within the noisy ICA framework given access to an algorithm (such as PEGI) to recover the directions of the columns of $A$. To this end, we first discuss the SINR optimal demixing solution within any decomposition of the ICA model into signal and noise as $\mathbf{X} = A\mathbf{S} + \boldsymbol{\eta}$. We then demonstrate that the SINR optimal demixing matrix is actually the same across all possible model decompositions, and that it can be recovered. The results in this section hold in greater generality than in section 2. They hold even if $m \geq n$ (the underdetermined setting) and even if the additive noise $\boldsymbol{\eta}$ is non-Gaussian.

Consider $B$ an $m \times n$ demixing matrix, and define $\hat{\mathbf{S}}(B) := B\mathbf{X}$ the resulting approximation to $\mathbf{S}$. It will also be convenient to estimate the source signal $\mathbf{S}$ one coordinate at a time: Given a row vector $\mathbf{b}$, we define $\hat{S}(\mathbf{b}) := \mathbf{b}\mathbf{X}$. If $\mathbf{b} = B_{k\cdot}$ (the $k^{\text{th}}$ row of $B$), then $\hat{S}(\mathbf{b}) = [\hat{\mathbf{S}}(B)]_k = \hat{S}_k(B)$ is our estimate to the $k^{\text{th}}$ latent signal $S_k$. Within a specific ICA model $\mathbf{X} = A\mathbf{S} + \boldsymbol{\eta}$, signal to intereference-plus-noise ratio (SINR) is defined by the following equation:

$$\text{SINR}_k(\mathbf{b}) := \frac{\text{var}(\mathbf{b}A_k S_k)}{\text{var}(\mathbf{b}A\mathbf{S} - \mathbf{b}A_k S_k) + \text{var}(\mathbf{b}\boldsymbol{\eta})} = \frac{\text{var}(\mathbf{b}A_k S_k)}{\text{var}(\mathbf{b}A\mathbf{X}) - \text{var}(\mathbf{b}A_k S_k)} . \tag{4}$$

$\text{SINR}_k$ is the variance of the contribution of $k^{\text{th}}$ source divided by the variance of the noise and interference contributions within the signal.

Given access to the mixing matrix $A$, we define $B_{\text{opt}} = A^H(AA^H + \text{cov}(\boldsymbol{\eta}))^\dagger$. Since $\text{cov}(\mathbf{X}) = AA^H + \text{cov}(\boldsymbol{\eta})$, it follows that $B_{\text{opt}} = A^H \text{cov}(\mathbf{X})^\dagger$. Here, $\text{cov}(\mathbf{X})^\dagger$ may be estimated from data, but due to the ambiguities of the noisy ICA model, $A$ (and specifically its column norms) cannot be.

Koldovský and Tichavský [15] observed that when $\boldsymbol{\eta}$ is a white Gaussian noise, $B_{\text{opt}}$ jointly maximizes $\text{SINR}_k$ for each $k \in [m]$, i.e., $\text{SINR}_k$ takes on its maximal value at $(B_{\text{opt}})_{k\cdot}$. We generalize this result in Proposition 2 below to include arbitrary non-spherical, potentially non-Gaussian noise.

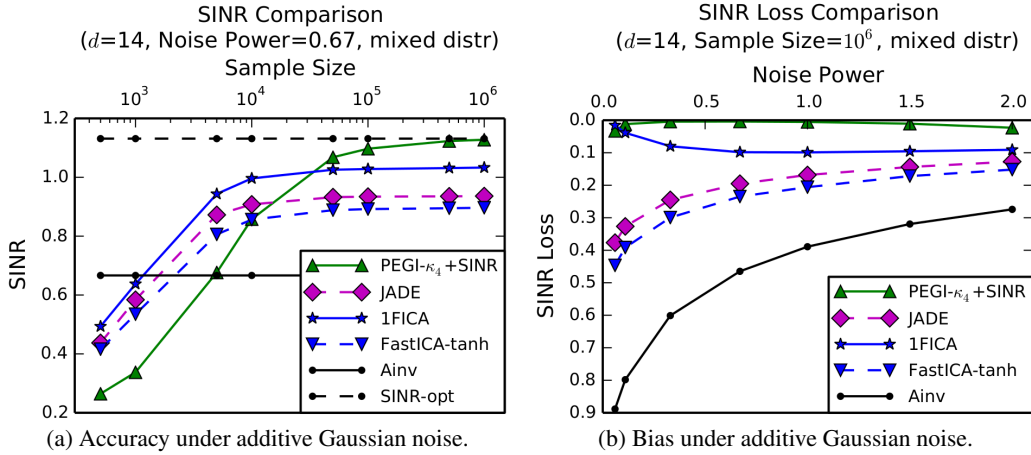

(a) Accuracy under additive Gaussian noise.  (b) Bias under additive Gaussian noise.

Figure 1: SINR performance comparison of ICA algorithms.

It is interesting to note that even after the data is whitened, i.e. $\mathrm{cov}(X) = \mathcal{I}$, the optimal SINR solution is different from the optimal solution in the noiseless case unless $A$ is an orthogonal matrix, i.e. $A^\dagger = A^H$. This is generally not the case, even if $\boldsymbol{\eta}$ is white Gaussian noise.

**Proposition 2.** *For each $k \in [m]$, $(B_{\mathrm{opt}})_{k\cdot}$ is a maximizer of $\mathrm{SINR}_k$.*

The proof of Proposition 2 can be found in the supplementary material.

Since SINR is scale invariant, Proposition 2 implies that any matrix of the form $DB_{\mathrm{opt}} = DA^H \mathrm{cov}(\mathbf{X})^\dagger$ where $D$ is a diagonal scaling matrix (with non-zero diagonal entries) is an SINR-optimal demixing matrix. More formally, we have the following result.

**Theorem 3.** *Let $\tilde{A}$ be an $n \times m$ matrix containing the columns of $A$ up to scale and an arbitrary permutation. Then, $(\tilde{A}^H \mathrm{cov}(X)^\dagger)_{\pi(k)\cdot}$ is a maximizer of $\mathrm{SINR}_k$.*

By Theorem 3, given access to a matrix $\tilde{A}$ which recovers the directions of the columns of $A$, then $\tilde{A}^H \mathrm{cov}(X)^\dagger$ is the SINR-optimal demixing matrix. For ICA in the presence of Gaussian noise, the directions of the columns of $A$ are well defined simply from $\mathbf{X}$, that is, the directions of the columns of $A$ do not depend on the decomposition of $\mathbf{X}$ into signal and noise (see the discussion in section 1.1 on ICA indeterminacies). The problem of SINR optimal demixing is thus well defined for ICA in the presence of Gaussian noise, and the SINR optimal demixing matrix can be estimated from data without any additional assumptions on the magnitude of the noise in the data.

Finally, we note that in the noise-free case, the SINR-optimal source recovery simplifies to be $\tilde{A}^\dagger$.

**Corollary 4.** *Suppose that $\mathbf{X} = A\mathbf{S}$ is a noise free (possibly underdetermined) ICA model. Suppose that $\tilde{A} \in \mathbb{R}^{n \times m}$ contains the columns of $A$ up to scale and permutation, i.e., there exists diagonal matrix $D$ with non-zero entries and a permutation matrix $\Pi$ such that $\tilde{A} = AD\Pi$. Then $\tilde{A}^\dagger$ is an SINR-optimal demixing matrix.*

Corollary 4 is consistent with known beamforming results. In particular, it is known that $A^\dagger$ is optimal (in terms of minimum mean squared error) for underdetermined ICA [19, section 3B].

## 4 Experimental Results

We compare the proposed PEGI algorithm with existing ICA algorithms. In addition to qorth+GI-ICA (i.e., GI-ICA with quasi-orthogonalization for preprocessing), we use the following baselines:
**JADE** [3] is a popular fourth cumulant based ICA algorithm designed for the noise free setting. We use the implementation of Cardoso and Souloumiac [5].
**FastICA** [12] is a popular ICA algorithm designed for the noise free setting based on a deflationary approach of recovering one component at a time. We use the implementation of Gävert et al. [10].
**1FICA** [16, 17] is a variation of FastICA with the tanh contrast function designed to have low bias for performing SINR-optimal beamforming in the presence of Gaussian noise.
**Ainv** performs oracle demixing algorithm which uses $A^\dagger$ as the demixing matrix.
**SINR-opt** performs oracle demixing using $A^H \mathrm{cov}(\mathbf{X})^\dagger$ to achieve SINR-optimal demixing.

We compare these algorithms on simulated data with $n = m$. We constructed mixing matrices $A$ with condition number 3 via a reverse singular value decomposition ($A = U\Lambda V^T$). The matrices $U$ and $V$ were random orthogonal matrices, and $\Lambda$ was chosen to have 1 as its minimum and 3 as its maximum singular values, with the intermediate singular values chosen uniformly at random. We drew data from a noisy ICA model $\mathbf{X} = A\mathbf{S} + \boldsymbol{\eta}$ where $\text{cov}(\boldsymbol{\eta}) = \Sigma$ was chosen to be malaligned with $\text{cov}(A\mathbf{S}) = AA^T$. We set $\Sigma = p(10\mathcal{I} - AA^T)$ where $p$ is a constant defining the *noise power*. It can be shown that $p = \frac{\max_{\mathbf{v}} \text{var}(\mathbf{v}^T \boldsymbol{\eta})}{\max_{\mathbf{v}} \text{var}(\mathbf{v}^T A\mathbf{S})}$ is the ratio of the maximum directional noise variance to the maximum directional signal variance. We generated 100 matrices $A$ for our experiments with 100 corresponding ICA data sets for each sample size and noise power. When reporting results, we apply each algorithm to each of the 100 data sets for the corresponding sample size and noise power and we report the mean performance. The source distributions used in our ICA experiments were the Laplace and Bernoulli distribution with parameters 0.05 and 0.5 respectively, the $t$-distribution with 3 and 5 degrees of freedom respectively, the exponential distribution, and the uniform distribution. Each distribution was normalized to have unit variance, and the distributions were each used twice to create 14-dimensional data. We compare the algorithms using either SINR or the SINR loss from the optimal demixing matrix (defined by SINR Loss = [Optimal SINR − Achieved SINR]).

In Figure 1b, we compare our proposed ICA algorithm with various ICA algorithms for signal recovery. In the PEGI-$\kappa_4$+SINR algorithm, we use PEGI-$\kappa_4$ to estimate $A$, and then perform demixing using the resulting estimate of $A^H \text{cov}(\mathbf{X})^{-1}$, the formula for SINR-optimal demixing. It is apparent that when given sufficient samples, PEGI-$\kappa_4$+SINR provides the best SINR demixing. JADE, FastICA-tanh, and 1FICA each have a bias in the presence of additive Gaussian noise which keeps them from being SINR-optimal even when given many samples.

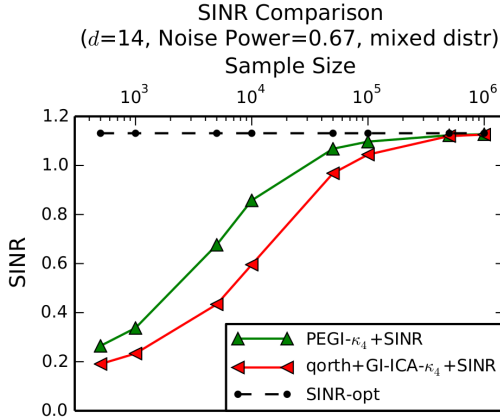

In Figure 1a, we compare algorithms at various sample sizes. The PEGI-$\kappa_4$+SINR algorithm relies more heavily on accurate estimates of fourth order statistics than JADE, and the FastICA-tanh and 1FICA algorithms do not require the estimation of fourth order statistics. For this reason, PEGI-$\kappa_4$+SINR requires more samples than the other algorithms in order to be run accurately. However, once sufficient samples are taken, PEGI-$\kappa_4$+SINR outperforms the other algorithms including 1FICA, which is designed to have low SINR bias. We also note that while not reported in order to avoid clutter, the kurtosis-based FastICA performed very similarly to FastICA-tanh in our experiments.

Figure 2: Accuracy comparison of PEGI using pseudo-inner product spaces and GI-ICA using quasi-orthogonalization.

In order to avoid clutter, we did not include qorth+GI-ICA-$\kappa_4$+SINR (the SINR optimal demixing estimate constructed using qorth+GI-ICA-$\kappa_4$ to estimate $A$) in the figures 1b and 1a. It is also assymptotically unbiased in estimating the directions of the columns of $A$, and similar conclusions could be drawn using qorth+GI-ICA-$\kappa_4$ in place of PEGI-$\kappa_4$. However, in Figure 2, we see that PEGI-$\kappa_4$+SINR requires fewer samples than qorth+GI-ICA-$\kappa_4$+SINR to achieve good performance. This is particularly highlighted in the medium sample regime.

**On the Performance of Traditional ICA Algorithms for Noisy ICA.** An interesting observation [first made in 15] is that the popular noise free ICA algorithms JADE and FastICA perform reasonably well in the noisy setting. In Figures 1b and 1a, they significantly outperform demixing using $A^{-1}$ for source recovery. It turns out that this may be explained by a shared preprocessing step. Both JADE and FastICA rely on a whitening preprocessing step in which the data are linearly transformed to have identity covariance. It can be shown in the noise free setting that after whitening, the mixing matrix $A$ is a rotation matrix. These algorithms proceed by recovering an orthogonal matrix $\tilde{A}$ to approximate the true mixing matrix $A$. Demixing is performed using $\tilde{A}^{-1} = \tilde{A}^H$. Since the data is white (has identity covariance), then the demixing matrix $\tilde{A}^H = \tilde{A}^H \text{cov}(\mathbf{X})^{-1}$ is an estimate of the SINR-optimal demixing matrix. Nevertheless, the traditional ICA algorithms give a biased estimate of $A$ under additive Gaussian noise.

## Footnotes

[1] $A^{-1}$ can be replaced with $A^\dagger$ ($A$'s pseudoinverse) in the discussion below for over-determined ICA.

[2]Alternatively, one may place the scaling information in the signals by setting $\|A_k\| = 1$ for each $k$.

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
