[Supplementary Material · pegi-long.pdf]

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

By Theorem 3, given access to a matrix $\tilde{A}$ which recovers the directions of the columns of $A$, then $\tilde{A}^H \mathrm{cov}(X)^\dagger$ is the SINR-optimal demixing matrix. For ICA in the presence of Gaussian noise, the directions of the columns of $A$ are well defined simply from $\mathbf{X}$, that is, the directions of the columns of $A$ do not depend on the decomposition of $\mathbf{X}$ into signal and noise (see the discussion in section 1.1 on ICA indeterminacies). The problem of SINR optimal demixing is thus well defined for ICA in the presence of Gaussian noise, and the SINR optimal demixing matrix can be estimated from data without any additional assumptions on the magnitude of the noise in the data.

Finally, we note that in the noise-free case, the SINR-optimal source recovery simplifies to be $\tilde{A}^\dagger$.

**Corollary 4.** *Suppose that $\mathbf{X} = A\mathbf{S}$ is a noise free (possibly underdetermined) ICA model. Suppose that $\tilde{A} \in \mathbb{R}^{n \times m}$ contains the columns of $A$ up to scale and permutation, i.e., there exists diagonal matrix $D$ with non-zero entries and a permutation matrix $\Pi$ such that $\tilde{A} = AD\Pi$. Then $\tilde{A}^\dagger$ is an SINR-optimal demixing matrix.*

*Proof.* By Theorem 3, $(AD^{-1}\Pi)^H \mathrm{cov}(\mathbf{X})^\dagger$ is an SINR-optimal demixing matrix. Expanding, we obtain: $(AD^{-1}\Pi)^H \mathrm{cov}(\mathbf{X})^\dagger = \Pi^H D^{-1} A^H (AA^H)^\dagger = \Pi^H D^{-1} A^\dagger = (AD\Pi)^\dagger = \tilde{A}^\dagger$.  □

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

# A    PEGI for Complex Signals

In Section 2, we showed how to perform gradient iteration ICA within a pseudo-Euclidean inner product space. In this appendix, we show how this PEGI algorithm can be extended to include complex valued signals. For clarity, we repeat the entire PEGI algorithmic construction from Section 2 with the necessary modifications to handle the complex setting.

Throughout this appendix, we assume a noisy ICA model $\mathbf{X} = A\mathbf{S} + \boldsymbol{\eta}$ where $\boldsymbol{\eta}$ is an arbitrary Gaussian noise independent of $\mathbf{S}$. We also assume that $m \leq n$, that $m$ is known, and that the columns of $A$ are linearly dependent.

## A.1    Fourth Cumulants of Complex Variables

The gradient iteration relies on the properties of cumulants. We will focus on the fourth cumulant, though similar constructions may be given using other even order cumulants of higher order. We will use two versions of the fourth cumulant which capture slightly different fourth order information. For a zero-mean random variable $X$, they may be defined as $\kappa_4(X) := \mathbb{E}[X^4] - 3\mathbb{E}[X^2]^2$ and $\kappa_4^\star(X) := \mathbb{E}[X^2 X^{*2}] - 2\mathbb{E}[XX^*]^2 - \mathbb{E}[X^2]\mathbb{E}[X^{*2}]$. For real random variables, these two definitions are equivalent, and they come from two different conjugation schemes when constructing the fourth order cumulant [see 7, Chapter 5, Section 1.2]. However, in general, only $\kappa_4^\star$ is guaranteed to be real valued. The higher order cumulants have nice algebraic properties which make them useful for ICA:

1. (Independence) If $X$ and $Y$ are independent random variables, then $\kappa_4(X + Y) = \kappa_4(X) + \kappa_4(Y)$ and $\kappa_4^\star(X + Y) = \kappa_4(X + Y)$.

2. (Homogeneity) If $\alpha$ is a scalar, then $\kappa_4(\alpha X) = \alpha^4 \kappa_4(X)$ and $\kappa_4^\star(\alpha X) = |\alpha|^4 \kappa_4^\star(X)$.

3. (Vanishing Gaussians) If $X$ is normally distributed then $\kappa_4(X) = 0$ and $\kappa_4^\star(X) = 0$.

In this appendix, we consider a noisy ICA model $\mathbf{X} = A\mathbf{S} + \boldsymbol{\eta}$ where $\boldsymbol{\eta}$ is a $\mathbf{0}$-mean (possibly complex) Gaussian and independent of $\mathbf{S}$. We consider the following functions defined on the unit sphere: $f(\mathbf{u}) := \kappa_4(\langle \mathbf{X}, \mathbf{u} \rangle)$ and $f_\star(\mathbf{u}) := \kappa_4^\star(\langle \mathbf{X}, \mathbf{u} \rangle)$. Then, expanding using the above properties we obtain:

$$f(\mathbf{u}) = \kappa_4\left( \sum\nolimits_{k=1}^m \langle A_k, \mathbf{u} \rangle S_k + \langle \mathbf{u}, \boldsymbol{\eta} \rangle \right) = \sum\nolimits_{k=1}^m \langle A_k, \mathbf{u} \rangle^4 \kappa_4(S_k)$$

Using similar reasoning, it can be seen that $f_\star(\mathbf{u}) = \sum_{k=1}^m |\langle A_k, \mathbf{u} \rangle|^4 \kappa_4^\star(S_k)$.

It turns out that some slightly non-standard notions of derivatives are most useful in constructing the gradient iteration in the complex setting. We use real derivatives for the gradient and we use the complex Hessian. In particular, expanding $u_k = x_k + iy_k$, we use the gradient operator $\nabla := \sum_{k=1}^n \mathbf{e}_k \frac{\partial}{\partial x_k}$. We make use of the operators $\partial u_k := \frac{1}{2}\left(\frac{\partial}{\partial x_k} - i\frac{\partial}{\partial y_k}\right)$ and $\partial u_k^* := \frac{1}{2}\left(\frac{\partial}{\partial x_k} + i\frac{\partial}{\partial y_k}\right)$ to define $\mathcal{H} := \sum_{j=1}^n \sum_{k=1}^n \mathbf{e}_k \mathbf{e}_j^T \partial u_k \partial u_j^*$. Applying this version of the Hessian is different than using real derivatives as in the gradient operation.

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

 space limitations, we omit the proof of Theorem 5. However, its proof is very similar that of an analogous result for the GI-ICA algorithm [21, Theorem 4].

In practice, we test near convergence by testing if we are still making significant progress. In particular, for some predefined $\epsilon > 0$, if there exists a unit modulus constant $c_k$ such that $\|\mathbf{u}_k - c_k\mathbf{u}_{k-1}\| < \epsilon$, then we declare convergence achieved and return the result. We may determine $c_k$ using the following fact.

---

**Algorithm 4** Full ICA matrix recovery algorithm. Estimates and returns two matrices: (1) $\tilde{A}$ is the recovered mixing matrix for the noisy ICA model $\mathbf{X} = A\mathbf{S} + \boldsymbol{\eta}$, and (2) $\tilde{B}$ is a running estimate of $\tilde{A}^\dagger$.

---

1: **Inputs:** $C, \nabla f$
2: $\tilde{A} \leftarrow 0, \tilde{B} \leftarrow 0$
3: **for** $j \leftarrow 1$ to $m$ **do**
4:      Draw $\mathbf{u}$ uniformly at random from $S^{n-1}$.
5:      **repeat**
6:          $\mathbf{u} \leftarrow \mathbf{u} - \tilde{A}\tilde{B}\mathbf{u}$
7:          $\mathbf{u} \leftarrow \nabla f(C^{\dagger^*}\mathbf{u})/\|\nabla f(C^{\dagger^*}\mathbf{u})\|$.
8:      **until** Convergence (up to a unit modulus factor)
9:      $\tilde{A}_j \leftarrow \mathbf{u}$
10:     $\tilde{B}_{j\cdot} \leftarrow [C^\dagger A_j/((C^\dagger A_j)^T A_j)]^T$
11: **end for**
12: **return** $\tilde{A}, \tilde{B}$

---

*Fact* 6. Suppose that $\mathbf{u}$ and $\mathbf{v}$ are non-orthogonal unit modulus vectors. The expression $\|\mathbf{u} - e^{i\theta}\mathbf{v}\|$ is minimized by the choice of $\theta = \text{atan2}(\text{Im}(\langle \mathbf{u}, \mathbf{v}\rangle), \text{Re}(\langle \mathbf{u}, \mathbf{v}\rangle))$.

Letting $\theta = \text{atan2}(\text{Im}(\langle \mathbf{u}_k, \mathbf{u}_{k-1}\rangle), \text{Re}(\langle \mathbf{u}_k, \mathbf{u}_{k-1}\rangle))$, we exit the loop if $\|\mathbf{u}_k - e^{i\theta}\mathbf{u}_{k-1}\| < \epsilon$.

### A.3 Full ICA Recovery Via the Pseudo-Euclidean GI-Update

We are able to recover a single column of $A$ in noisy ICA. However, for full matrix recovery, we would like (given recovered columns $A_{\ell_1}, \ldots, A_{\ell_j}$) to be able to recover a column $A_k$ such that $k \notin \{\ell_1, \ldots, \ell_j\}$ on demand.

The main idea behind the simultaneous recovery of all columns of $A$ is two-fold. First, instead of just finding columns of $A$ using Algorithm 3, we simultaneously find rows of $A^\dagger$. Then, using the recovered columns of $A$ and rows of $A^\dagger$, we may project $\mathbf{u}$ onto the orthogonal complement of the recovered columns of $A$ within the $\langle \cdot, \cdot\rangle_C$ pseudo-Euclidean inner product space.

**Recovering rows of $A^\dagger$.** Suppose we have access to a column $A_k$ (which may be achieved using Algorithm 3). Let $A_{k\cdot}^\dagger$ denote the $k^{\text{th}}$ row of $A^\dagger$. Then, we note that $C^\dagger A_k^* = (A^*DA^T)^\dagger A_k^* = d_{kk}^{-1}(A^T)_k^\dagger = d_{kk}^{-1}(A_{k\cdot}^\dagger)^T$ recovers $A_{k\cdot}^\dagger$ up to an arbitrary, unknown constant $d_{kk}^{-1}$. However, the constant $d_{kk}^{-1}$ may be recovered by noting that $\langle A_k, A_k\rangle_C = (C^\dagger A_k)^T A_k = d_{kk}^{-1}$. As such, we may estimate $A_{k\cdot}^\dagger$ as $[C^\dagger A_k/((C^\dagger A_k)^T A_k)]^T$.

**Enforcing Orthogonality During the GI Update.** Given access to $\mathbf{u} = \sum_{k=1}^m \alpha_k(\mathbf{u})A_k + P_{A^\perp}\mathbf{u}$, some recovered columns $A_{\ell_1}, \ldots, A_{\ell_r}$, and corresponding rows of $A^\dagger$, we may zero out the components of $\mathbf{u}$ corresponding to the recovered columns of $A$. Letting $\mathbf{u}' = \mathbf{u} - \sum_{j=1}^r A_{\ell_j} A_{\ell_j\cdot}^\dagger \mathbf{u}$, then $\mathbf{u}' = \sum_{k\in[m]\setminus\{\ell_1,\ldots,\ell_r\}} \alpha_k(\mathbf{u})A_k + P_{A^\perp}\mathbf{u}$. In particular, $\mathbf{u}'$ is orthogonal (in the $\langle \cdot, \cdot\rangle_C$ space) to the previously recovered columns of $A$. This allows us to modify the non-orthogonal gradient iteration algorithm to recover a new column of $A$.

Using these ideas, we obtain the Algorithm 4 for recovery of the ICA mixing matrix. Within this Algorithm, step 6 enforces orthogonality with previously found columns of $A$, guaranteeing that convergence is to a new column of $A$.

**Practical Construction of $C$** We suggest the choice of $C = \frac{1}{4}\sum_{k=1}^n \mathcal{H}f_\star(\mathbf{e}_k)$, as it can be shown from equation (6) that $\sum_{k=1}^n \mathcal{H}f_\star(\mathbf{e}_k) = A^*DA^T$ with $d_{kk} = \|A_k\|^2 \kappa_4^\star(S_k)$. This deterministically guarantees that each latent signal has a significant contribution to $C$.

## B   Proof of Proposition 2

*Proof.* This proof is based on the connection between two notions of optimality, minimum mean squared error and SINR. The mean squared error of the recovered signal $\hat{S}(\mathbf{b})$ from $k^{\text{th}}$ latent signal is defined as $\text{MSE}_k(\mathbf{b}) := \mathbb{E}[|S_k - \hat{S}(\mathbf{b})|^2]$. It has been shown [14, equation 39] that $B_{\text{opt}}$ jointly

minimizes the mean squared errors of the recovered signals. In particular, if $\mathbf{b} = (B_{\text{opt}})_{k\cdot}$, then $\mathbf{b}$ is a minimizer of $\text{MSE}_k(\mathbf{b})$.

We will first show that finding a matrix $B$ which minimizes the mean squared error has the side effect of maximizing the magnitude of the Pearson correlations $\rho_{S_k, \hat{S}_k(B)}$ for each $k \in [m]$, where $\rho_{S_k, \hat{S}_k(B)} := \frac{\mathbb{E}[S_k \hat{S}_k^*(B)]}{\sigma_{S_k} \sigma_{\hat{S}_k(B)}}$. We will then demonstrate that if $B$ is a maximizer of $|\rho_{S_k, \hat{S}_k(B)}|$, then $B_{k\cdot}$ is a maximizer of $\text{SINR}_k$. These two facts imply the desired result. We will use the convention that $\rho_{S_k, \hat{S}_k(B)}$ is 0 if $\sigma_{\hat{S}_k(B)} = 0$.

We fix a $k \in [m]$. We have:
$$\text{MSE}_k(\mathbf{b}) = \mathbb{E}[S_k S_k^* - 2\operatorname{Re}(S_k \hat{S}^*(\mathbf{b})) + \hat{S}(\mathbf{b})\hat{S}^*(\mathbf{b})]$$
$$= 1 - 2\sigma_{\hat{S}(\mathbf{b})}\operatorname{Re}(\rho_{S_k, \hat{S}(\mathbf{b})}) + \sigma_{\hat{S}(\mathbf{b})}^2 .$$

Letting $\omega = \operatorname{sgn}(\rho_{S_k, \hat{S}(\mathbf{b})})$, we obtain
$$\rho_{S_k, \hat{S}(\omega\mathbf{b})} = \frac{\mathbb{E}[S_k \hat{S}^*(\omega\mathbf{b})]}{\sigma_{S_k}\sigma_{\hat{S}(\omega\mathbf{b})}} = \omega^* \frac{\mathbb{E}[S_k \hat{S}^*(\mathbf{b})]}{\sigma_{S_k}\sigma_{\hat{S}(\mathbf{b})}} = |\rho_{S_k, \hat{S}(\mathbf{b})}| . \tag{8}$$

Further, $\text{MSE}_k(\omega\mathbf{b}) = 1 - 2\sigma_{\hat{S}(\mathbf{b})}|\rho_{S_k, \hat{S}(\mathbf{b})}| + \sigma_{\hat{S}(\mathbf{b})}^2 \leq \text{MSE}_k(\mathbf{b})$ with equality if and only if $\rho_{S_k, \hat{S}(\mathbf{b})}$ is real and non-negative. As such, all global minima of $\text{MSE}_k$ are contained in the set $\mathcal{A} = \{\mathbf{b} \mid \rho_{S_k, \hat{S}(\mathbf{b})} \in [0, 1]\}$, and we may restrict our investigation to this set.

We define a function $g(x, y) := 1 - 2xy + y^2$ such that under the change of variable $x(\mathbf{b}) = \sigma_{\hat{S}(\mathbf{b})}$ and $y(\mathbf{b}) = \rho_{S_k, \hat{S}(\mathbf{b})}$, we obtain $\text{MSE}_k(\mathbf{b}) = g(x, y)$. Let $M = \max_{\mathbf{b} \in \mathcal{A}} \rho_{S_k, \hat{S}(\mathbf{b})}$ and let $y_0 \in [0, M]$ be fixed. Then, $\arg\min_{x \in \mathbb{R}} g(x, y_0) = y_0$ with the resulting value $g(y_0, y_0) = 1 - y_0^2$. As such, the minimum of $g(x, y)$ over the domain $\mathbb{R} \times [0, M]$ occurs when $x = y = M$. If $M = 0$, then the choice of $\boldsymbol{\xi} = \mathbf{0}$ satisfies that $x(\boldsymbol{\xi}) = y(\boldsymbol{\xi}) = 0$, making $\text{MSE}_k(\boldsymbol{\xi}) = g(x, y)$ the global minimum of $\text{MSE}_k$. If $M \neq 0$, then we may choose $\boldsymbol{\xi}$ such that $y(\boldsymbol{\xi}) = \rho_{S_k, \hat{S}(\boldsymbol{\xi})} = M$. As $\sigma_{\hat{S}(\boldsymbol{\xi})} > 0$ must hold, it follows that there exists $\alpha \in (0, \infty)$ such that setting $\boldsymbol{\zeta} = \alpha\boldsymbol{\xi}$, we obtain $(\sigma_{\hat{S}(\boldsymbol{\zeta})} =)x(\boldsymbol{\zeta}) = y(\boldsymbol{\xi})$. Since $y(\mathbf{b}) = \rho_{S_k, \hat{S}(\mathbf{b})}$ is scale invariant, we obtain that $x(\boldsymbol{\zeta}) = y(\boldsymbol{\zeta}) = y(\boldsymbol{\xi}) = M$, making $\boldsymbol{\zeta}$ a global minimum of $\text{MSE}_k$. In both cases, it follows that if $\mathbf{b}$ minimizes $\text{MSE}_k(\mathbf{b})$, then $\mathbf{b}$ maximizes $\rho_{S_k, \hat{S}(\mathbf{b})}$ over $\mathcal{A}$.

From equation (8), we see that $\max_{\mathbf{b} \in \mathbb{C}^n} |\rho_{S_k, \hat{S}(\mathbf{b})}| = \max_{\mathbf{b} \in \mathcal{A}} \rho_{S_k, \hat{S}(\mathbf{b})}$. Thus if $\mathbf{b}$ is a minimizer of $\text{MSE}_k(\omega\mathbf{b})$, then $\mathbf{b}$ is also a maximizer of $|\rho_{S_k, \hat{S}(\mathbf{b})}|$ as claimed.

We now demonstrate that $\mathbf{b}$ is a maximizer of $|\rho_{S_k, \hat{S}(\mathbf{b})}|$ if and only if it is also a maximizer of $\text{SINR}_k(\mathbf{b})$. Under the conventions that $\frac{x}{0} = +\infty$ when $x > 0$ and that $\frac{0}{0} = s\infty$ where $s = -1$ for maximization problems and $s = +1$ for minimization problems, the following problems have equivalent solution sets over choices of $\mathbf{b}$:

$$\max_{\mathbf{b}} \text{SINR}_k(\mathbf{b}) \equiv \max_{\mathbf{b}} \frac{\mathbb{E}[|\mathbf{b}A_k S_k|^2]}{\operatorname{var}(\hat{S}(\mathbf{b})) - \mathbb{E}[|\mathbf{b}A_k S_k|^2]} \equiv \max_{\mathbf{b}} \frac{|\mathbb{E}[S_k \hat{S}^*(\mathbf{b})]|^2}{\operatorname{var}(\hat{S}(\mathbf{b})) - |\mathbb{E}[S_k \hat{S}^*(\mathbf{b})]|^2}$$
$$\equiv \min_{\mathbf{b}} \frac{\operatorname{var}(\hat{S}(\mathbf{b})) - |\mathbb{E}[S_k \hat{S}^*(\mathbf{b})]|^2}{|\mathbb{E}[S_k \hat{S}^*(\mathbf{b})]|^2} \equiv \min_{\mathbf{b}} \frac{\operatorname{var}(\hat{S}(\mathbf{b}))}{|\mathbb{E}[S_k \hat{S}^*(\mathbf{b})]|^2}$$
$$\equiv \max_{\mathbf{b}} \frac{|\mathbb{E}[S_k \hat{S}^*(\mathbf{b})]|^2}{\operatorname{var}(\hat{S}(\mathbf{b}))} \equiv \max_{\mathbf{b}} |\rho_{S_k, \hat{S}(\mathbf{b})}|^2 .$$

In the above, the first equivalence is a rewriting of equation (4). To see the second equivalence, we note that $|\mathbb{E}[S_k \hat{S}^*(\mathbf{b})]|^2 = |\mathbb{E}[S_k (\mathbf{b}A\mathbf{S} + \mathbf{b}\boldsymbol{\eta})^*]|^2 = |\mathbf{b}A_k|^2$ using the independence of $S_k$ from all other terms. Then, noting that $|\mathbf{b}A_k|^2 = \mathbb{E}[|\mathbf{b}A_k S_k|^2]$ gives the equivalence. The fourth equivalence is only changing the problem by the additive constant $-1$. $\qquad\square$