[Reviews · NeurIPS 2015]

Submitted by Assigned_Reviewer_1

The paper proposes a new algorithm for noisy ICA. It uses only fourth-order cumulant, thus achieving (asymptotically) unbiased estimation for Gaussian noise.

The paper is well positioned and motivated and seems to be of high technical quality.

One drawback is that the novelty as well as improvement in performance w.r.t. previous work in [21] may be limited. But perhaps the improvement is useful, at least the ensuing algorithm is surprisingly simple.

Another drawback is that the number of data points needed to demonstrate any utility of the method is large: tens of thousands in a 14-dimensional setting. This may mean that the method is not likely to be very useful in practice.
Summary: An interesting new algorithm for noisy ICA, with theory that looks sound. But perhaps limited novelty, and modest improvement performance in simulations.

Submitted by Assigned_Reviewer_2

This paper addresses the problem of blind signal extraction (recovery) in the presence of Gaussian noise.

To solve the noisy ICA problem the authors employ a fixed point iteration to achieve a maximization of directional kurtosis.

The fixed-point iteration is carried out in a pseudo-Euclidean space.

Based on the proposed "pseudo-Euclidean gradient iteration (PEGI)" two algorithms are presented: one for estimating a single column of the mixing matrix and a second one using an adapted deflation method to find the full mixing matrix while making sure not to find the same component again.

clarity:

The paper is clearly written and a pleasure to read. However some key references of related work are missing (see below).

originality:

The approach is similar to the well-known fastICA

method and the GI-ICA [21] (Voss et al.,NIPS 2013), however it differs from those methods in an important aspect: It does neither rely on the pre-whitening (orthogonalization) step nor on "quasi-whithening" (quasi-orthogonalization).

The main contributions

(1)

gradient-based fixed point iterations for kurtosis maximisation are done in a pseudo-Euclidean space

(2) analysis and insight how to choose this pseudo-Euclidean space optimally for the noisy ICA case

are technically sound and sufficiently novel.

Significance:

Performing ICA in the noise-corrupted setting is an important problem of potential interest to the NIPS crowd.

Though the robustness to Gaussian noise is inherent to all higher-order cumulant-based approaches and is well know, the proposed method to integrate this into a fixed-point algorithm is admittedly elegant and thus the contribution can be considered significant.

A further plus is the straightforward extension to the complex case (which is shown in supplemental material)

The experimental section is a bit weak, only synthetic data is considered and many other robust ICA methods are not included in comparison.

At least experimental results demonstrating improvements over GI-ICA indicate that the cumbersome quasi-orthogonalization as in [21] is no longer needed.

On the other hand there exist many methods for the noisy ICA problem. In particular, the algorithm in [R2] "RobustICA" would be the real competitor, because as shown in [R3] the fixed-point algorithm can be seen as a special case of gradient descent with an optimally selected stepsize as proposed in [R2] but such a comparison is not included. Also the relative Newton method of Zibulevsky [R5] for minimization of a quasi-ML cost function would have provided a better baseline.

A further issue is that the estimation of 4-th order cumulants / kurtosis in general needs a LOT of data (1e+5--1e+6 samples are often needed, as can be seen from figure 2)

Furthermore the practical usefulness for real-world data remains to be demonstrated.

further comments:

-It would be recommended to compare fastICA with pow3 not tanh nonlinearity in Fig1

-Why is the iteration index k omitted in algorithm 2 ?

additional references:

[R1] Sergio Cruces, A. Cichocki, S. Amari, "The Minimum Entropy and Cumulant Based Contrast Functions for Blind Source Extraction", Lecture Notes in Computer Science LNCS-2085, Springer-Verlag, pp. 786-793, 2001.

[R2] Vicente Zarzoso and Pierre Comon, "Robust Independent Component Analysis by Iterative Maximization of the Kurtosis Contrast With Algebraic Optimal Step Size" IEEE TRANSACTIONS ON NEURAL NETWORKS, VOL. 21, NO. 2, FEBRUARY 2010,10.1109/TNN.2009.2035920

[R3] H. Li and T. Adali. A class of complex ICA algorithms based on the kurtosis cost function.

IEEE Transactions on Neural Networks, 19(3):408-420, 2008.

[R4] Javidi S, Mandic DP, Took CC, and Cichocki A: "Kurtosis-based blind source extraction of complex non-circular signals with application in EEG artifact removal in real-time.", Front Neurosci, 5, 105 (2011)

[R5] M. Zibulevsky,"Relative Newton and Smoothing Multiplier Optimization Methods for Blind Source Separation", chapter in the book: S. Makino, T.W. Lee and H. Sawada eds., Blind Speech Separation, Springer Series: Signals and Communication Technology XV, 2007

[R6] P. A. Regalia and E. Kofidis, "Monotonic convergence of fixed-point algorithms for ICA," IEEE Trans. Neural Netw., vol. 14, no. 4, pp. 943-949, Jul. 2003.

summary

The authors present an elegant modification of the fixed-point iteration for robust ICA in a kurtosis-maximization framework.

The provided insights would be of interest to the ICA community even though the practical usefulness of the method for real-world data remains to be demonstrated.

Summary: The authors present an elegant modification of the fixed-point iteration for robust ICA in a kurtosis-maximization framework.

The provided insights would be of interest to the ICA community even though the practical usefulness of the method for real-world data remains to be demonstrated.

Submitted by Assigned_Reviewer_3

The paper proposes an algorithm called PEGI which is suitable for the noisy ICA problem. The algorithm is based on a gradient iteration in a pseudo-Euclidian space where the inner product is based on the Hessian of the contrast function (ie. the kurtosis). The authors show that with probability 1 the algorithm extracts the normalized columns of the mixing matrix. Then they show that the extracted mixing matrix estimate can be used to make an SINR-optimal recovery of the sources.

The disadvantage of the method is that it needs a large number of samples N (in the example of Fig 1a, over 50,000 samples) in order to outperform standard ICA methods such as JADE, FastICA, or 1FICA.

Moreover, compared to the existing GI-ICA method [21] it is not siginificantly superior for large data sets (Fig 2 shows marginal improvement over GI-ICA for N>50,000).

The presentation of the paper is quite clear and the proposed method includes originality. Moreover the theoretical results regarding convergence and the optimal SINR reconstruction theorem count towards the positive points of the paper. However, the limitations mentioned above weaken the significance of the method. Perhaps the authors could strengthen it by identifying specific signal or noise settings where PEGI clearly outperforms standard methods like FastICA or JADE. Otherwise it can be yet another run-of-the-mill method within the multitude of methods performing ICA.

Summary: A pseudo-Euclidian space algorithm suitable for noisy ICA. Clearly presented and original. However the performance improvements with respect to existing methods is not very convincing raising the risk of this being yet another ICA method.

Submitted by Assigned_Reviewer_4

This paper deals with the problem of noisy ICA and proposes an extension of the Gradient-Iteration (GI) ICA algorithm proposed by Voss et al. Since the conventional GI-ICA algorithm is based on an assumption that the columns of A are orthogonal, it requires a rather complicated preprocessing step called the quasi-orthogonalization to linearly transform the data to make columns of A orthogonal. The key innovation of the proposed method is to sidestep this process by formulating the signal recovery problem as a fixed point method in an indefinite inner product (pseudo-Euclidean) space.

Although a line of the ICA research is well introduced in Section 1, the research on underdetermined blind signal separation (BSS) should also be briefly mentioned. By appending one column and one row in A and S, the noisy ICA model X = AS + n can be equivalently rewritten in the form X = A'S' where A' and S' denote the appended matrices. Practically, existing methods for underdetermined BSS can also be used to solve the signal recovery problem under this setting where the noise is treated as a source. In this regard, literature on underdetermined BSS are also somewhat relevant to this work, which I think should not be completely disregarded.

This paper seems to reuse many sentences from the following manuscript, which can downloaded from the internet: J. Voss, M. Belkin, L. Rademacher, "Optimal Recovery in Noisy ICA," 2015. To avoid any risk of being accused of plagiarism or self-plagiarism, I suggest the authors avoid using exactly the same sentences as much as possible.

It should be mentioned somewhere that $^\dagger$ denotes the pseudo-inverse.

A concluding section is missing. Please include it in the resubmitted version.

Quality: Very good Clarity: Excellent Originality: Good Significance: Good
Summary: This paper deals with the problem of noisy ICA and proposes an extension of Gradient-Iteration (GI) ICA algorithm proposed by Voss et al. The key innovation of the proposed method is to sidestep the quasi-orthogonalization step by formulating the signal recovery problem as a fixed point method in an indefinite inner product (pseudo-Euclidean) space.

Submitted by Assigned_Reviewer_5

This paper extends a recently proposed noisy ICA method called GI-ICA. The new method called PEGI may avoid the quasi-orthogonalization preprocessing step necessary in GI-ICA. The authors also give a general optimality result related to source recovery in noisy ICA.

Overall, the paper is well written with high technical quality. The method (PEGI) is a reasonable and relevant extension of GI-ICA, and the result shows the method works reasonably. However most materials in Section 2 are from the GI-ICA paper and the modification to GI-ICA is relatively straightforward. In this regard, the contribution of this paper itself may be limited.

The additional optimality result (Section 3) is interesting in its own right but it looks quite fundamental so I'm skeptical about the novelty. At least a similar result can be found in: A. F. Naguib, "Adaptive Antennas for CDMA Wireless Network," Ph.D. Dissertation, Stanford Univ., Stanford, CA, 1996. (Chap.3), which seems to be an oft-cited reference related to beamforming. I'm also wondering how it could be related to source recovery based on MAP estimation, which is popularly used in probabilistic noisy ICA methods.

The presentation of the method needs some improvements in some specific points. First, at the first reading it was unclear to me how the gradient and hessian of kurtosis (eqs. 1 and 2) could be estimated from data. I found the detail in the GI-ICA paper [21] but the present paper should also describe or appropriately refer to the existing result. Second, it was difficult to figure out why the quasi-orthogonalization is problematic while PEGI is not, even though they both rely on the same matrix C. Some more detailed explanation on this point should be necessary to show the advantage (and motivation) of the new method more clearly.

Summary: The method is reasonable and relevant extension of previously proposed GI-ICA method while the contribution of this paper itself seems relatively limited. The presentation should be improved in some specific points.

Author Feedback
Author rebuttal: We thank all reviewers for the helpful comments and suggestions. Below we respond to some specific points.

Reviewer_1

"One drawback is that the novelty as well as improvement w.r.t. previous work in [21] may be limited."

We show improvement over the algorithm from [21] at all sample sizes in fig. 2. In addition, our algorithm is much simpler conceptually and in implementation.

Reviewer_4

"performance improvements with respect to existing methods is not very convincing raising the risk of this being yet another ICA method"

Within the noisy ICA setting, there are fewer algorithms than within the noise free ICA setting, as there are more technical difficulties to overcome. That tends to lead to algorithms which are (a) more complicated, (b) rely on randomness (i.e., Yeredor's method [22] and Goyal, et al [11]), and/or (c) have significant restrictions on the model, such as requiring same sign kurtosis (see FOOBI [4], BIOMe [1], and Arora, et al [2]).

PEGI provides a simple fixed point method which does not suffer from these restrictions and does not need complicated quasi-orthogonalization used in [21]. We hope that this algorithm will be of interest to the ICA community not only for performance reasons, but also on a conceptual level.

Reviewer_6

Thanks for the detailed review and helpful references and suggestions.

"Why is the iteration index k omitted in algorithm 2 ?"

Thanks for pointing this out. We will make the notation consistent between algorithms 1 and 2.

"It would be recommended to compare fastICA with pow3 not tanh nonlinearity in Fig1"

In our experiments, FastICA-pow3 performed very similarly to FastICA-tanh so we did not include it to avoid cluttering the plots. We will mention it in the revision.

Reviewer_7

Thanks for the helpful suggestions.

"underdetermined blind signal separation (BSS) should also be briefly mentioned"

Several of the algorithms that we mention in the related works section (BIOMe, FOOBI, and Goyal, et al's method) are actually underdetermined algorithms. We will clarify this in the revision.

Reviewer_9

3. "Did not see the point of requiring that A had to have independent columns. If I understood it right, full rank would be enough."

In PEGI, A is assumed to have at most as many columns as rows. In this setting, it is equivalent to say either that A is full rank or that its columns are linearly independent.

6. "Finally, PEGI has a cubic nonlinearity. Although FastICA with "tanh" has better theoretical properties, it would be interesting to know (even if only in text) how the cubic nonlinearity in FastICA faired in the comparison (not expecting it to be much different from tanh, though)."

In our experiments, FastICA with both pow3 and tanh performed very similarly. We will state this in the revision.

Reviewer_10.

"However most materials in Section 2 are from the GI-ICA paper and the modification to GI-ICA is relatively straightforward.
...
Second, it was difficult to figure out why the quasi-orthogonalization is problematic while PEGI is not, even though they both rely on the same matrix C."

Matrix C in PEGI is not the same as matrix C in GI-ICA. To be precise, matrix C in PEGI is matrix M in GI-ICA. GI-ICA then additionally eigen-decomposes M and uses it in another operation involving the 4th cumulant to obtain the quasi-orthogonalization matrix. This makes the algorithm in GI-ICA less stable than PEGI. The matrices are different and they are used in different ways. In particular, matrix C in PEGI is not positive definite and its use is only possible thanks to one of the algorithmic innovations of the current paper (indefinite inner product). Perhaps our notation is confusing, we will clarify that.

"The additional optimality result (Section 3) is interesting in its own right but it looks quite fundamental so I'm skeptical about the novelty"

We note in the introduction to the paper that SINR-optimal demixing appears in the array sensor systems literature and cite [6] for a closely related result. On the other hand, there is some confusion in the ICA literature on this point. The ICA papers of which we are aware that explicitly discuss MSE and SINR optimal demixing do not make this observation. In fact, they propose different, more limited, strategies for addressing SINR optimality. For instance, [14] discusses how SINR optimal demixing can be approximated with extra sensors, and the discussion in [16] assumes low noise variance within a fixed ICA model. We hope that our discussion in the context of ICA (including the proof of Theorem 3) will be helpful in clarifying these issues.

"it was unclear to me how the gradient and hessian of kurtosis (eqs. 1 and 2) could be estimated from data. I found the detail in the GI-ICA paper [21] but the present paper should also describe or appropriately refer to the existing result."

Good point. We will add an appropriate reference/explanation.